# The First Whole Genome Sequence Discovery of the Devastating Fungus *Arthrinium rasikravindrae*

**DOI:** 10.3390/jof8030255

**Published:** 2022-03-02

**Authors:** Abdul Qayoom Majeedano, Jie Chen, Tianhui Zhu, Shujiang Li, Zeeshan Ghulam Nabi Gishkori, Sumbul Mureed Mastoi, Gang Wang

**Affiliations:** 1College of Forestry, Sichuan Agricultural University, Chengdu 611130, China; 2019504004@stu.sicau.edu.cn (A.Q.M.); B20172701@edu.sicau.cn (J.C.); 14087@sicau.edu.cn (S.L.); smmastoi@sau.edu.pk (S.M.M.); james@stu.sicau.edu.cn (G.W.); 2Department of Plant Pathology, College Agriculture and Biotechnology, Zhejiang University, Hangzhou 310058, China; 12116150@zju.edu.cn

**Keywords:** first, report, whole-genome sequence, *Arthrinium rasikravindrae*, devastating, fungus

## Abstract

Devastating fungi are one of the most important biotic factors associated with numerous infectious diseases not only in plants but in animals and humans too. *Arthrinium rasikravindrae* a devastating fungus is responsible for severe infections in a large number of host plants all over the world. In the present study, we analyzed the whole genome sequence of devastating fungus *A. rasikravindrae* strain AQZ-20, using Illumina Technology from the Novogene Bio-informatics Co., Ltd. Beijing, China. To identify associated annotation results, various corresponding functional annotations databases were utilized. The genome size was 48.24 MB with an N90 (scaffolds) length of 2,184,859 bp and encoded putative genes were 11,101, respectively. In addition, we evaluated the comparative genomic analyses with 4 fungal strains of Ascomycetes. Two related species showed a strong correlation while others exhibited a weak correlation with the *A. rasikravindrae* AQZ-20 fungus. This study is a discovery of the genome-scale assembly, as well as annotation for *A. rasikravindrae*. The results obtained from the whole genome sequencing and genomic resources developed in this study will contribute significantly to genetic improvement applications against diseases caused by *A. rasikravindrae*. In addition, the phylogenetic tree, followed by genomic RNA, transcriptomic, proteomic, metabolic, as well as pathogenic data reported in current research will provide deep insight for further studies in the future.

## 1. Introduction

Fungi are the most destructive and diverse groups of pathogenic fungi responsible for many lethal diseases in plants, especially those belonging to the family Poaceae and Cyperaceae. Such fungi exist in nature as endo-parasites and are widely distributed all over the world [1,2]. As per the National Centre for Biotechnology Information (NCBI), the taxonomic classification of this fungus is under Fungi, Dikarya, Ascomycota, Saccharomyces, Pezizomycotina, Leotiomyceta, Sordariomyceta, Sordariomycetes, Xylariomycetidae, Xylariales, Apiosporaceae, Arthrinium, *A. rasikravindrae* and was first reported by Singh et al. [3] from Norway. Previously, *A. rasikravindrae* fungus was reported as *A. phaeospermum* and then re-determined as *A. rasikravindrii* but recently, it has orthographically been corrected to *A. rasikravindrae*. The current name is now widely recognized as well as accepted and also used in Mycobank (http://www.mycobank.org/ assessed on 16 August 2020) and Index Fungorum (http://www.indexfunorum.org assessed on 16 August 2020) [3,4]. Numerous associations were originating dead culms of *Ph. aurea* from Europe as well by [5]. To date, 31 different species of this genus have been reported from all over the world [6]. The data available in NCBI GenBank indicates that 20 entries have been mentioned so far based on the diseased samples collected from different hosts and geographical regions globally including Brazil, China, India, Japan, Netherlands, Norway, Thailand, and Switzerland, [7]. Many studies from China have shown that there exists a rich species diversity of Arthrinium in the bamboo (*Bambusa vulgaris*) plant [4,8]. Out of the total 31 Arthrinium species, 17 have been reported from host plants [4,8,9,10]. These hosts are widely distributed in nature and have never been comprehensively surveyed [11]. To date, 32 species of Arthrinium fungi have been reported [6], The Arthrinium genus inhabit a large number of hosts and substrates, including air, lichens, marine algae, plants, soil debris, and even human tissues [9,11].

This genus can be associated with phytopathogens, such as *A. ovatum* and *A. hydei* from Hong Kong, *A. kogelbergense* from South Africa, *A. malaysianum* and *A. pseudospegazzinii* from Malaysia, *A. phragmites* from Italy, *A. pseudosinense* from the Netherlands, *A. xenocordella* in Zimbabwe, *A. arundinis* in America, [12], *A. sacchari* in Canada and *A. phaeospermum* in China, respectively [12,13]. Based on the recent studies conducted on *A. phaeospermum*, only two cases have been noticed where *A. phaeospermum* acted as a causal agent of infectious diseases in human beings [14,15,16], however, many others are known to produce diverse bioactive compounds with a variety of pharmacological applications [17,18], In addition, some fungal species associated with genus Arthrinium are known to produce some toxic compounds that inhibit the disease-causing potential of a broad range of human pathogens including filamentous fungi, bacteria and yeasts [19]. The study of Li et al. [14] has found that *A. phaeospermum* produces growth-promoting substances in *C. kobomugi* [20]. Previous studies [21,22] have shown that a non-protein toxic compound mediates the physiological, biochemical, metabolic activities, and pathogenicity of *A. phaeospermum*, and is hence responsible for the devastation in hybrid bamboo tissues. Additionally, *A. phaeospermum* has also been reported to synthesize protein-based toxic compounds as well [23,24,25,26] and hybrid bamboo cells have been reported to contain a constricting position on the semi-permeable membrane [27]. It was also shown that proteinaceous toxic compounds damage the mitochondrial membrane, increase lipid peroxidation of mitochondria, disrupt membrane integrity, as well as inhibit respiration [28]. This shows the evidence that toxin protein played an essential part in the pathogenicity of *A. phaeospermum* [29]. Metabolic reports have illustrated that *A. phaeospermum* is involved in the synthesis of numerous biochemical compounds including tetrahydroxy anthraquinone pigment, succinic acid, ergosterol, phenols (C_18_O_5_) [30], gibberellic acid [31] 3-nitropropionic acid [32], and arthrichitin (C_33_H_46_N_4_O_9_), respectively [33]. For instance, the study of Shrestha et al. [34,35] revealed that *A. phaeospermum* possesses various bioactive enzymes that degrade plant cell wall constituents, including exo-cellulase, endo-nuclease, xylanase, ligninase, and β-glucosidase, which perform imperative functions in the host plant cell wall degradation [14]. However, no studies have reported beneficial or dangerous effects of *A. rasikravindrae* on living organisms. To date, the complete genome sequences of 178 human and 24 plant-related fungi have been obtained [14]. Out of these, various devastating pathogens including *Ustilago maydis* in 2003 [36], the draft genome sequence of *Alternaria alternate* in 2015 [37], *Candida auris* in 2021, *Penicillium digitatum* [38], *Verticelium albo-atrum* [39], *Candida albicans* [40], *Arthrinium phaeospermum* AP-Z13, and *Diporthe capsici* in 2020 have been successfully sequenced [14,41].

Previous studies have reported that various organisms have been utilized for genomic transformations as individuals or cells, such as the Cancer Genome Association and *Arabidopsis* Genome Project [42]. Significant efforts have been made in the past five years to improve assemblies of genomes, and the improvement of nearly complete genomes using long-read sequencing technologies [43]. Simultaneously, there are no whole genomic studies have been found of *A. rasikravindrae* worldwide.

The objectives of the current research are (1) to search the whole genome sequence of the *A. rasikravindrae* strain AQZ-20, (2) to evaluate the species-specific and extended gene families by performing comparative genomic analysis with the four Xylariales fungal strains classified under Ascomycetes. Our findings will provide enhanced insights for future detailed studies of *A. rasikravindrae*.

## 2. Materials and Methods

### 2.1. Fungal Material

In this research, the *A. rasikravindrae* strain was isolated from the bamboo (*Lignania intermedia*) shoots given by the Key Laboratory of Forest Conservation, Sichuan Agricultural University, Chengdu, China. Pure culture of *A. rasikravindrae* strain was inoculated on (PDA) medium in 90-mm Petri dishes and cultivated for seven days at 25 °C. Later, the collected fresh mycelia samples were sent to the NovaSeq Illumina PE150 Novogene Bioinformatics Technology Co. Ltd. Beijing to extract the DNA and construct the whole genome sequence of the strain *A. rasikravindrae* AQZ-20. Strain identifiers of comparative studies for each isolate, accession numbers to the genomes, Species, Naming in GenBank, Total Length (MB), GC (%), N50, Scaffolds, and Contigs are listed in Appendix A. The sequencing metrics for each genome were obtained from NCBI.

### 2.2. Extraction of Genome DNA, Library Construction, Sequencing, and Assembly

Extraction of DNA from the mycelia samples was performed using the SDS method [44]. The purity and integrity of the DNA was observed by Agarose gel-electrophoresis, while Qubit was utilized for quantification. Later, an assemblage for (SMRT) sequencing was created with an interruption size of 20 kb using the SMRTbell™ Template Prep kit, as well as its attribution was measured on the Qubit 2.0 Fluorometer (Thermo Scientific, Buildings 3 & 6 & 7, No. 27, Xinjinqiao Road, Pudong New Area, Shanghai, China). The size of the interruption was resolved using Agilent 2100 (Agilent Technologies, No. 3 Wangjing North Road, Chaoyang District, Beijing, China). Lastly, the whole genome of *A. rasikravindrae* was sequenced in exploitation by the PacBio Sequel platform [45]. The SMRT Link [41,46,47] was utilized to accomplish the construction of reads as well as to acquire the explorative consequences. The redundant sequences in the preliminary assembly outcomes were separated, as well as the reads were matched with the assembled sequence. By deceiving the GC content of the assembled sequence as well as the extended depth of reads, the GC bias, as well as recurrent sequences of the genome, were figured, as well as the part with assembly mistakes were chastised to acquire the best assembly results [41].

### 2.3. Gene Prediction and Genome Assembly

The whole-genome information for *A. rasikravindrae* was obtained using AUGUSTUS [48] as well as the homologous GeneWise software, based on the homologous protein sequence of *A. phaeospermum* (taken from NCBI) as the remarkable sequence. The two outcomes were then incorporated with EVM and confirmed using PASA in the second round to acquire the reasoning consequences for the coding genes. These aggregations were then used to screen the assembled genomes for repeats using Repeat Masker (RM) version open-4.0.7 [49], RM was utilized to predict the dispersed repeats sequence (DRs). Tandem Repeats (TR) Finder [50], was utilized to predict the TR sequence (TRs). For ribosomal RNA (rRNA) prevision, all the genome was investigated against rRNA sequences, while for the transfer RNA (tRNA), (tRNAs) genes were determined utilizing the tRNAscan-SEv 2.0 [51], as well as rRNAmmer [52] which was utilized to anticipate rRNAs, sRNAs, and in advance, microRNA (miRNA); small nucleolar (snRNA) genes were predicted using BLAST against the Rfam database [53,54]. 

The predictive genes protein sequences were equivalenced and functional annotations were assigned to the genes using functional databases, including (KOG), [55], (GO), (KEGG) database [56,57], NR Protein Database (NR), [58], P450, Pfam, Swiss-Prot, [59] and (TCDB), [60], were also used for functional annotation results. A whole-genome Blast [61] was executed against the preceding seven databases and the secretory proteins were foreseen using the SignalP [62] database whereas secondary metabolism gene clusters were predicted using antiSMASH [62]. Furthermore, (PHI) and fungal virulence factor database (DFVF) were utilized to examine and confirm the pathogenicity of the pathogens [63]. Carbohydrate-active enzymes were expected using the Carbohydrate-Active enzymes Database [64]. The genome sequence of strain *A. rasikravindrae* AQZ-20 was deposited in the GenBank database with the following accession numbers: JACVVL000000000. The assembly and sequenced genome raw data reported in this paper are subordinated with NCBI BioProject: PRJNA661692 and BioSample: SAMN16069862 within GenBank. The SRA accession number is SRR12768822. All the deposited data will be published online after article publication.

### 2.4. Comparative Genomic Analysis

To compare the genome data sets, the genome sequence of *Arthrinium phaeospermum* (accession number: QYRS00000000.1) [14] was used as the credited genome for *A. rasikravindrae* homological taxons. The genomes of *Arthrinium malaysianum* STlab-iicb (accession number: (QUSE00000000.1), *Fusarium oxysporum* FO2 [35] (accession number: (AAXH00000000.1) [65], and *Fusarium proliferatum* ET1 (accession number: (FJOF00000000), a pathogenic fungal species responsible for causing similar infections were used for comparison with genomic information obtained for *A. rasikravindrae*. The alignment of the genome among the representative genome as well as cited genome was evaluated using the MUMmer [66] while LASTZ [67,68,69] was utilized to conduct pairwise comparisons between *A. rasikravindrae* and the four reference genomes. The gene family was constructed with multiple software: Blast [70] and collinearity between the genomes was performed by MCScan X software, [71], whereas MEGA software was used for phylogenetic analysis, and the phylogenetic tree was built by the Neighbour- Joining method [9].

## 3. Results

### 3.1. Assembly of the Genome and Characteristics of Genome Analysis

The material used for whole-genome analysis was isolated from bamboo shoots. The whole-genome analysis of the *Arthrinium rasikravindrae* was carried out and the number of reads in raw offline information was obtained. The preliminary assembly results showed that *A. rasikravindrae* contained 33 contigs and the maximum length of the assembled contig was noticed as 4,830,709 bp. The N50 contig length was 2,356,085 bp and total contig length was 45,879,981 bp. Then, reads were straightened to the assembly sequence and the ultimate assembly results after alignment and assessment also showed similar results as noticed in the preliminary assembly. The assembled contig GC content was 52.66% (Table 1). 

### 3.2. Gene Prediction

The analysis of genomic data predicted a total of 11,101 coding genes with an entire length of 45,874,955 bp, accounting for 31.73% of the total genome length. The coding genes mean length was noticed as 1311 bp (Table 2). Furthermore, we analyzed the genomic data for dispersed repeat sequences (DRs) as summarized in Table 2. The DRs were classified into six categories namely LTR, DNA, LINE, SINE, RC, and Unknown which accounted for 0.14%, 0.09%, 0.08%, 0.01%, 0.007% and 0.001% of the total genome with average length of 81, 74, 89, 96, 78 and 71 base pairs, respectively. The tandem repeats sequences (TRs), Minisatellite DNA, and Microsatellite DNA were recorded for 0.61%, 0.43% and 0.08% of the total genome with a repeat size of 1~456 bp, 10~60 bp, and 2~6 bp, respectively (Table 3). In addition, we also noticed RNAs in *A. rasikravindrae* as summarized in Table 4. The highest number of RNAs were recorded as tRNA (268), followed by snRNA (15), sRNA (2), and mi RNA (1), respectively.

### 3.3. Gene Annotation

To further verify the 11,101 coding genes and functional characterization, the gene annotation analysis was carried out using different databases including CAZy.CAZy is a database of Carbohydrate-Active enZYmes (CAZymes), DFVF, GO (Gene Ontology), KEGG (Kyoto Encyclopedia of Genes and Genomes), KOG (Eukaryotic Orthologous Groups of proteins), NR (Nuclear Receptor), P450, Pfam, PHI (The Pathogen-Host Interactions database (PHI-base), SwissProt, (http://www.expasy.ch/sprot, assessed on 4 September 2021), and TCDB (The Transporter Classification Database), respectively.

The gene annotation by the GO platform annotated 6792 coding genes in *A. rasikravindrae*, as categorized into three major classes, i.e., molecular function (11 sub-classes), cellular component (13 sub-classes), and biological process (25 sub-classes) as shown in Figure 1. Among the three classes, six functional entries were noticed with more than 2000 annotated genes. These entries included cellular process (3162 genes), metabolic process (3526 genes), cell (2351), cell part (2351), binding (3302), and catalytic activities (3560), respectively.

To further classify the function of identified coding genes in *A. rasikravindrae*, the KEGG pathway annotation was performed. The KEGG database annotated 8893 genes and classification graph of KEGG metabolic pathways has been shown in Figure 2. The biological pathways are divided into six categories, each of which is subdivided, and each category is labeled with the relevant information. Among all the categories, the highest number of annotated genes were noticed in metabolic pathways (2650), within which the carbohydrate metabolism exhibited the most abundant number of genes.

KOG database annotated about 2085 genes of functional categories based on the protein functions (Figure 3). The protein functions are primarily centered in the amino acid transport and metabolism (164 genes), energy production and conversion (179 genes), general function diction only (242 genes), post translation modification, protein turnover, and chaperones (249 genes), translation, ribosomal structure and biology (214 genes), and other aspects, respectively.

### 3.4. NR, Pfam, and Swiss-Prot Annotation Database

The NR database annotation is summarized in Appendix A. Our results showed the top 20 species that exhibited matching genes with the annotated genes of *A. rasikravindrae*. The highest number of genes were matched with *Pseudornasarlella vexala* (2526), followed by *Pestalotiopsis fici* (1978) and *Hypoxylon* sp. (1041), respectively. However, the species *Colletotrichum incanum* showed the lowest number of matching genes, i.e., 32 in the NR annotation database. In addition, 197 genes were matched in fungal sp. (Appendix A). The number of annotated genes in the Pfam and Swiss-prot database were 6792 and 3241, respectively.

### 3.5. TCBD Function Classification

We employed the TCBD platform for functional classification of transporters in *A. rasikravindrae* (Appendix A). Transporters canister transport certain virulence factors or compounds concealed by pathogenic fungi once they contaminate hosts so that causal agents can contaminate particular plants more effortlessly; consequently, they are similarly deliberated as a class of pathogenic aspects in pathogenic fungi. The results showed that there were 539 genes annotated in the TCDB database and grouped in nine categories as shown in Appendix A. The highest number of transporter genes were noted in electrochemical potential driven transporters (197 genes), monitored by primary active transporters (151), accessory factors involved in transport (33), transmembrane electron carriers (3), channels/pools (81), group transporters (5), and incompletely characterized transport systems (69), respectively (Appendix A).

### 3.6. The Carbohydrate Enzyme Classification and Annotation

The CAZy database was employed to explore the carbohydrate enzyme classification and annotation in the genome of *A. rasikravindrae* (Appendix A). This database mapped 624 genes and these genes were further grouped into six CAZy classes. The highest number of genes (309) were allocated in GH (glycoside hydrolases) class, followed by the CAZy class oxidoreductase Auxiliary Activities (AA), carbohydrates Carbohydrate-Binding Module (CBM), Glycosyltransferases (GT), Carbohydrate Esterases (CE), and Polysaccharide Lyase (PL), respectively, in which 113, 112, 107, 58 and 10 genes were assigned.

### 3.7. Secreted Proteins Prediction

Typically, many pathogenesis-related and unrelated secreted proteins can be predicted in genomes using bioinformatics and computer-based prediction algorithms, which help to elucidate the molecular mechanisms of pathogen-plant interactions. A total of 1326 signal peptide protein structures, 2271 proteins with transmembrane structures, and 1081 secreted proteins were predicted in the genomic data of *Arthrinium rasikravindrae*.

### 3.8. P450

The fungal cytochrome P450 database (FCPD) is a superfamily of enzymes involved in the production and modification of different compounds. The annotation results for the cytochrome (Figure 4). Most (98) of the genes were originated to encode the E-class cyt P450, group I.

A total of 8731 P450 genes were isolated from 113 fungi and oomycetes. According to the position of genes in the database, they were divided into 16 groups, which were aggregated into 2579 by triple MCL Class. P450 enzymes contributed to the construction of vital metabolites in organisms but play an important role in adapting to dissimilar environments via the modification of harmful compounds too. Using blast software, the amino acid sequence of the target species was compared with the P450 database, and the gene of the target species and its corresponding functional annotation was assigned to obtain the annotation results (Figure 4). Since each sequence alignment result may be more than one, to ensure its biological significance, an optimal ratio was reserved for annotation of the gene. The blast results are in the format of M8. At the same time, the annotated results of some databases are also provided. In addition, the P450 gene exhibits conservative sequence characteristics, namely the heme binding domain motif CG and the oxygen binding motif Er of the k-helix. According to these two sequence characteristics, the P450 gene was found, and finally, the P450 gene sequence based on blast and sequence characteristics was provided. Unless otherwise specified, the sample names in the following contents represent the target species (Figure 4).

### 3.9. Gene Clusters of Secondary Metabolites

The gene clusters for secondary metabolites were predicted. There were 47 genes and four clusters encoding (NRPS), as well as 82 genes and eight clusters in T1PKS. In addition, 13 genes and one cluster in NRPS, T1PKS (NRPS, Non-Ribosomal Peptide Synthetase Cluster; T1PKS, Type I Pks (PolyKetide Synthase)), while 15 genes and one cluster in T1PKS, Indole were noticed. Furthermore, our results also revealed 32 genes and five clusters in terpene and 26 genes and two clusters in NRPS-like, respectively (Appendix A).

### 3.10. Pathogen Host Interaction (PHI)

There are 1415 genes annotated in the PHI database. The assigned genes for pathogenicity of *A. rasikravindrae* in the pathogen host interaction (PHI) database were classified into 29 categories (Appendix A). Results revealed that the maximum (520) number of matched genes was related to the functional class of “unaffected pathogenicity”. The second highest functional class (496) belongs functional class of “reduced virulence”. In addition, 130 matched genes were in the NA function class, 105 matched genes were found in the loss of pathogenicity, 71 in the lethal function class, and 20 in the GO function class, respectively (Appendix A). Amongst them, the resemblance of A02487 and A03535 genes of *A. rasikravindrae* and between CAM as well as GzGPA1 genes of *Magnaporthe oryzae* was 100% and 99.7%, respectively (Appendix A).

### 3.11. DFVF

DFVF database is an inclusive database of recognized fungal virulence issues. It collects 2058 pathogenic genes released by 228 fungal strains from 85 genera. In the present genomic study, there were 453 genes in DFVF, 3.85% predicted genes were accounted. Among them, 99.7% similarity was found between the A02487 gene of *A. rasikravindrae* and the A03535 gene of *Fusarium oxysporum*.

### 3.12. The Circular Whole-Genome Map of A. rasikravindrae

The outermost loop is the situation coordinates of the genome sequence, from the outer to the inner side, respectively, the GC content of the genome is counted by the window (genome/1000) bp and the step size (genome/1000) bp. The blue part inward designated that the whole genome sequence GC content of this portion is poorer than the average GC content, and the purple part outwards is the conflict. The high peak value indicates the superior difference from the average GC content. Genome GC skews value: window (genome/1000) bp, step size (genome/1000) bp, the particular algorithm is GC/G + C. The innermost green portion shows that the GC content of the section is inferior to the C content, and the outer pink portion is contradictory. The gene density (in the window genome/1000 bp, step size genome/1000 bp, respectively, counts the gene thickness of coding genes, rRNA snRNA tRNA, the darker the color, the greater the gene mass in the window), and chromosome duplication.

### 3.13. Comparative Genomic Analysis

#### 3.13.1. Analysis of Collinearity

The genomic analysis of collinearity between *A. rasikravindrae* and *A. phaeospermum AP-Z13*, *A. malaysianum STlab-iicb*, *F. proliferatum ET1*, and *F. oxysporum Fo2* genomes is shown in Figure 5a–d. The collinearity analysis showed the highest collinear correlation between *A. rasikravindrae* and *A. phaeospermum*, followed by *A. rasikravindrae* and *A. malaysianum*. For instance, the collinearity analysis showed that there were 11,040 collinearity blocks in *A. rasikravindrae* and *A. phaeospermum,* and the entire base length (35,846,460 bp) in the collinear block of *A. phaeopsermum* was elucidated for 73.99% of the entire gene length (48,449,939 bp) presenting great collinear correlation between the two fungal species. Similarly, 8988 collinearity blocks in *A. rasikravindrae* and *A. malaysianum* and the entire base length (23,555,373 bp) in the collinear block of *A. malaysianum* was elucidated for 51.09% of the entire gene length (46,110,019 bp), representing the second great collinear correlation amongst *A. rasikravindrae* and *A. malaysianum*. However, the direction of collinearity was very low regarding the remaining two fungal species, i.e., *F. oxysporum* and *F. proliferatum*. The collinear block number within *A. rasikravindrae* and *F. oxysporum* and *F. proliferatum* was 1740 and 1161, respectively, and the whole base length was elucidated for 2.92% as well as 1.77% of the entire gene length.

#### 3.13.2. The Core and Pan Genomic Analysis

The core genes are a set of genes with direct involvement in the disease pathway, while the pan-genes are the entire set of genes within a clade. The number of specific genes in *A. rasikravindrae* and *A. phaeospermum* was found to be 6337 and 8372, harmoniously, the core genes number was 2839, and the pan genes number was 427. The specific genes number in *A. malaysianum* was 8055, the core genes number was 3480, and the pan genes number was 520, respectively. The specific genes number in *F. oxysporum* Fo2 was 8492, the core genes number was 6809, and the pan genes number was 3396, respectively. The specific genes number in *F. proliferatum* was 11,158, the core genes number was 6935, and the pan genes number was 3438, respectively (Figure 6a).

#### 3.13.3. Analysis Gene Family

The gene families of *A. rasikravindrae*, *A. phaeospermum*, *A. malaysianum*, *F. oxysporum*, and *F. proliferatum* were subjected to cluster analysis. Cluster analysis was performed to find the unique genes family of core and pan-genome. The total and the unique number of gene families of *A. rasikravindrae* and other spp. were presented in the form of Venn diagrams. The specific genes number, the total gene family’s number, and the number of families’ unique genes are displayed in Figure 6b.

Based on the identity value of the database sequence and target sequence these are the 10 common genes shown in Figure 6a. The genes exhibiting identity values higher than 90 were A03535 (99.7%) unknown function, A11076 (99.7%), A10939 (97%) unidentified or un classified sequence, A02329 (95.8%) other sequence, A08957 (95.5%) unidentified or un classified sequence, A08380 (93.4%) synthetic construct, A00946 (93.3%) conceptual translation, A08495 (91.5%) unknown function, A07052 (91%) unknown function and A08159 (90.7%) synthetic construct, (Table 5.) respectively, in the DFVF database. The genes with identity values higher than 90 were A02487 (100%), A03535 (99.7%), A08794 (97.5%), A06181 (97.4%), A08957 (95.5%), A10642 (95.5%), A10939 (95.3%), A01584 (95.2%), A03803 (94.9%), A02448 (94.8%), A03859 (94.3%), A08892 (93.8%), A05643 (93.5%), A00274 (93.5%), A09431 (92.7%), A05678 (92.6%), A09487 (92.2%), A01337 (91.3%), A08159 (90.7%), A09292 (90.3%), A03055 (90.2%) and A01326 (90%), respectively, in PHI. 

## 4. Discussion

*A. rasikravindrae*, is a phytopathogenic fungus that provides a promising tool for epidemiological analysis and is expected to replace traditional methods in the near future. Whole genome sequencing is expected to revolutionize the surveillance and diagnosis of infectious diseases due to its high resolution [72]. The Pacbio RSII sequencing platform is an innovative third-generation sequencing platform at home and abroad. It has the advantages of long read length, high precision, and high sensitivity [73]. Our present study estimated the whole genome analysis and established that the entire genome size was 48.24-Mb (Figure 7). The SMRT technology directed by the PacBio platform can convey more extremely accurate long reads. Thus, we generated genome sequencing merging PacBio long-reads sequencing assembly. The sequencing data generated by the PacBio platform showed the contig N90 of 2,184,859 bp while encoded putative genes were 11,101. It can faultlessly collaborate with PacBio to whole high superiority genome assembly. Our study revealed 2839 core and 427 pan genes between *A. rasikravindrae* and *A. phaeospermum* as shown in (Figure 6a). All these findings verified that the PacBio platform assembly facilitated obtaining high-quality genome data. Our results are consistent with the studies of Chin et al. [74], and Huo et al. [75].

The entire genome of the *A. rasikravindrae* assembly was sequenced by this platform and matched with the genomes of *A. phaeospermum* AP-Z13, *A. malaysianum*, *F. oxysporum,* and *F. proleferatum*. The result showed the high similarity between *A. phaeospermum* and *A. rasikravindrae* of the same genus (Appendix A and Figure 8). Our results advise that *A. rasikravindrae* might be elaborate in the production of some secondary metabolites, cell wall degrading enzymes, and gene transcription regulation processes protease, which is closely associated with pathogenesis.

Improved genome assembly and exact gene collinearity ensure the inference of thousands of plausible homologs arising from evolutionary events, polyploidization, or speciation. Collinear homologs were likely to be generated simultaneously in corresponding events [43]. Our results showed 11,040 collinearity blocks in *A. rasikravindrae* and *A. phaeospermum* which resembled 73.99% of the entire gene length (48,449,939 bp), thus representing a large linear correlation between both fungal species. Similarly, the 8988 collinearity blocks in *A. rasikravindrae* and *A. malaysianum* resembled approximately 51.09% of the entire gene length (46,110,019 bp) as shown in Figure 5. 

From a taxonomic point of view, the current results obtained from the genome sequencing of *A. rasikravindrae* show a surprising side of the fungal kingdom and diversity among closely related species. However, *A. rasikravindrae* and *A. phaeospermum* both belong to class Sordariomycetes, hence similar methods are often utilized for detailed studies. Furthermore, whole-genome sequence studies revealed that the genes encoding the proteins were intact in both species *A. phaeospermum* AP-Z13 and *A. rasikravindrae* AQZ-20 of the same genus *Arthrinium* (Figure 8). Our findings suggest that the genes encoding proteins mainly include A03865, a gene located in the small nuclear ribonucleoprotein as well as A05515, A06181, and A02487 involved in putative protein, GTPase, actin cytoskeleton organization, cell polarity, and gene expression or gene silencing, respectively. 

Additionally, the *A. rasikravindrae* gene annotated in the NR database exhibited a higher gene identity with *P. vexata*, *M. oryzae* or *P.* fici and lower gene identity with *Colletotrichum incanum, Magnaporthiopsis poae,* and *Nectria heamatococca*, respectively (Appendix A). Previous studies reported that higher gene identity with *P. vexata*, *M. oryzae,* and *P. fici* can cause Fusarium ear blight in the host *Hordeum vulgare*. Among the NRPS gene clusters encoded by the *A. rasikravindrae* cluster, the largest number of *A. rasikravindrae* genes was paired with *Pseudornasarlella vexala* (Appendix A). In addition, studies have found that species associated with the genus *Arthrinium* are also the casual agents of dead bamboo columns disease in *Bambusa indica* [69]. 

Furthermore, the A02487 (Q9UWF0) and A03535 (I1RNF9) genes in *A. rasikravindrae* were paired with the CAM gene of *M. oryzae*, which caused rice blast and head blast in host *Triticum aestivum* [76], and the GzGPA1 gene of *Fusarium graminearum*, which caused Fusarium ear blight [77]. Therefore, the A02487 and A03535 genes might be associated with rice blast pathogenicity [9], and Fusarium ear blight in Barley. The GzGPA1 pathogenicity of the barley-directed strain was the same as that of the wild-type strain. GzGPA1 is a conserved low salt protein that encodes 353 (AA), highly similar to *A. nidulans* FadA in the genome database of *F. graminearum* (93% identity), and also negatively controls mycotoxin production. *F. graminearum* is the causative agent of small grain head blight disease and produces mycotoxins *deoxynivalenol* (DON) and *zearalenone* (ZEA) in diseased masses, hence its exposure is a danger to the health of both animals and humans. In *G. zeae* sexual expansion, GzGPA1 played a crucial role [76]. Deletion of GzGPA1 is somewhat valuable in PDA asexual expansion, by the evolutionary proportions of the modified being homologous on Barley wild type strain.

However, the similarity between the A02487 or A03535 genes of *A. rasikravindrae* and *F. oxysporum,* as well as the CAM and GzGPA1 genes of *Magnaporthe oryzae* was 100% and 99.7%, respectively. In *M. oryzae*, the homology between *A. rasikravindrae* and the A06181 gene was 97.4% in wheat and rice [76]. *A. rasikravindrae* is a common plant pathogen that induces actin cytoskeleton organization and cell polarity; establishment of appressorium and altered gene expression silencing, which are closely related to pathogenesis. However, four of the 10 core genes in the DFVF database have different functions, as shown in Table 5.

## 5. Conclusions

In the current research, the Pac bio RSII platform was utilized to complete the genome sequence of *A*. *rasikravindrae* strain AQZ-20, which revealed unique features in fungal genomes. The genomic study contained functional annotations, such as gene clusters of secondary metabolite synthesis, pathogen-host interaction database, and fungal virulence factors database to analyze preliminary pathogenicity analysis. In addition, the gene families were identified and the whole genome compilation results were determined as a final product. The whole-genome sequencing of *A. rasikravindrae* provided important data and the theoretical basis for transcriptomic, proteomic, and metabolic studies of this fungus and further enriched the *A. rasikravindrae* database showing specific association with strain AQZ-20. The results of gene analysis of *A. rasikravindrae* and genome sequence evaluation among several fungal species will contribute towards detailed further research about the fungal pathogenicity in forestry, medicine, and agriculture. Simultaneously, the preliminary comparison and analysis of several *Arthrinium* genomes provide a foundation to study the evolutionary system of *Arthrinium*, elucidate its pathogenesis, and support the prevention and control of diseases caused by the pathogenic fungi of *Arthrinium.* Moreover, this study will provide valuable resources for genomic studies and reference information for the management of different endophytic *Arthrinium* species as well as other related fungi.

## Figures and Tables

**Figure 1 jof-08-00255-f001:**
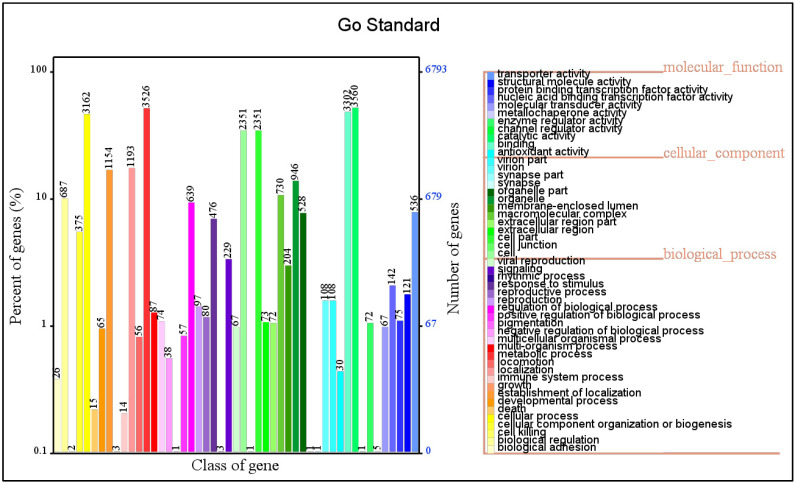
GO functional annotation of the *A. rasikravindrae* genome. In the (GO) annotation, the protein-coding sequences of the fungus *A. rasikravindrae* genome are divided into three major categories, (MF), (CC), and (Bprocess); different colors represent 49 sub-classes from these three major categories. Although in this figure the *x*-axis represents the class of genes and *y*-axis represents the percent of genes (%), while the *z*-axis represents the number of genes with a scale up to 6793.

**Figure 2 jof-08-00255-f002:**
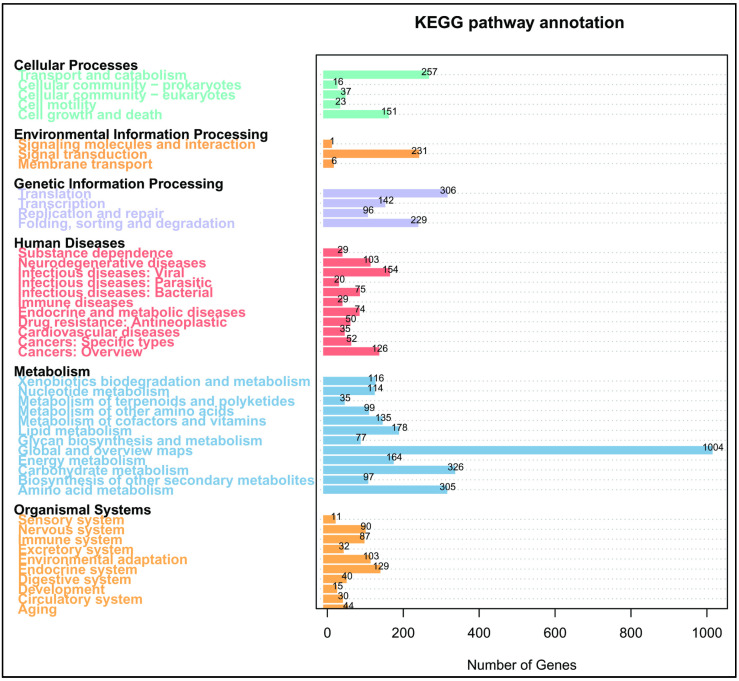
Represents the (KEGG) functional annotation of the *A. rasikravindrae* genome. In this figure, the major classes with their names and the number of genes from the concerned sub-class divisions are represented. The *x*-axis represents the scale for several genes up to 1000. KEGG functional annotation is divided into six major classes viz; Cellular process, Environmental information processing, Genetic information processing, Human diseases, Metabolism or Organismal system, and 40 subclasses. Each sub-class has been represented with a distinct color.

**Figure 3 jof-08-00255-f003:**
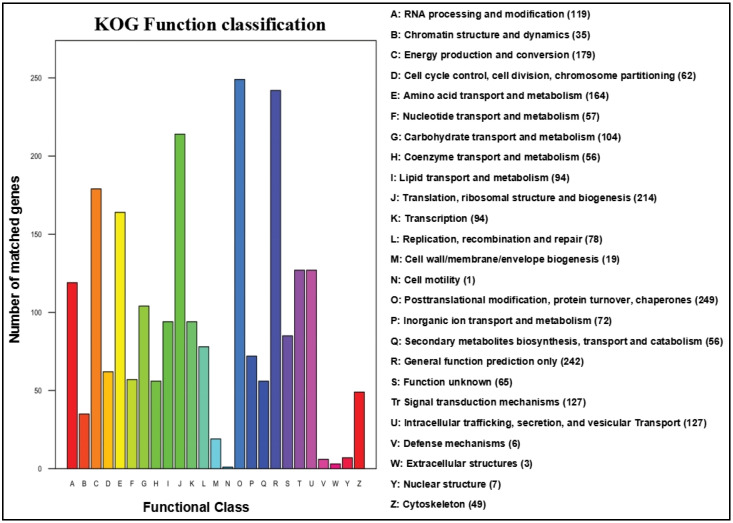
Clusters of orthologous groups of proteins (KOG) functional classification of proteins in *A. rasikravindrae*. In this figure, the *x*-axis shows the functions of the class and the *y*-axis shows the number of matched genes. Although KOG function classification is divided into 26 groups categorized by A–Z. Different colors with their names and the number of genes is also mentioned.

**Figure 4 jof-08-00255-f004:**
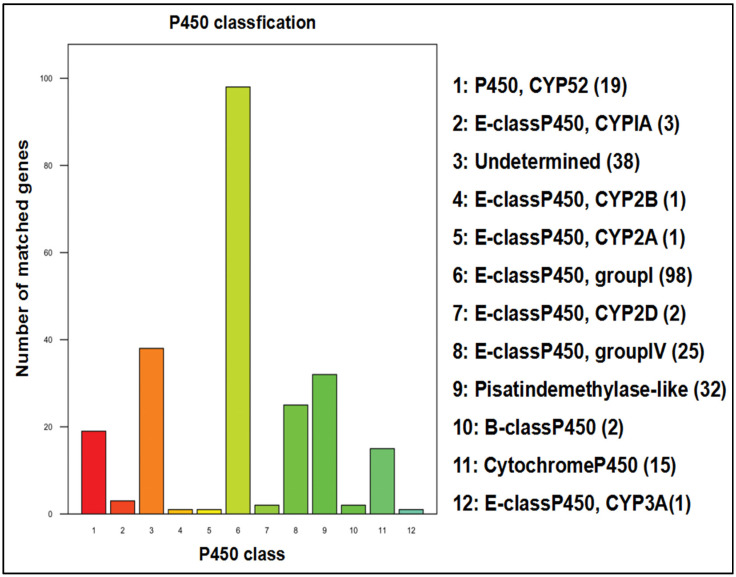
Functional classification of cytochrome P450 in the *A. rasikravindrae* genome. In this figure, the *x*-axis shows the P450 class, *y*-axis represents the number of matched genes with a scale from 0–100. P450 classification includes 12 classes.

**Figure 5 jof-08-00255-f005:**
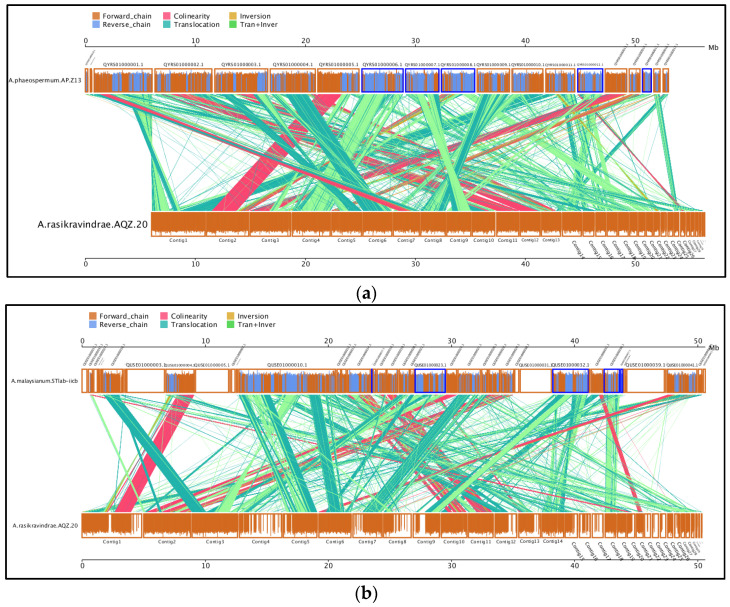
Parallel collinearity comparison analysis between *Arthrinium rasikravindrae* AQZ-20 and the *A. phaeospermum* AP-Z13, *A. malaysianum* STlab-iicb, *F. proliferatum* ET1, *F. oxysporum* Fo2, and *A. rasikravindrae* species. This figure shows the comparative genomic analysis between *A. rasikravindrae* and four other closely related fungus from the same genus Ascomycetes. In which two fungus are from the same family while the other two are from different families. (**a**) collinearity between *A. rasikravindrae* and *A. phaeospermum* strain AP-Z13, (**b**) collinearity between *A. rasikravindrae* and *A. malaysianum* STlab-iib, (**c**) collinearity between *A. rasikravindrae* and *F. proliferatum* ET1, and (**d**) collinearity between *A. rasikravindrae* and *F. oxysporum* Fo2 species. While at top left side of these figures (**a**–**d**) scale has been shown with different colors and corresponding names. The parallel lines with boxes upside and downside represent the contigs number while these are counted by the scale 0–60 MB upside the parallel lines.

**Figure 6 jof-08-00255-f006:**
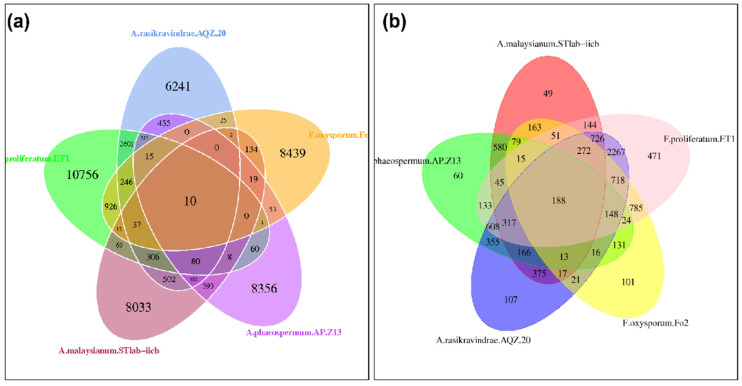
(**a**) Core and Pan Gene orthological Venn diagram of the AQZ-20, AP-Z13, STlab-iicb, ET1, and Fo2. The Venn map of *A. rasikravindrae* AQZ-20 was obtained according to the annotation results of the *A. phaeospermum* AP-Z13, *A. malaysianum* STlab-iicb, *F. oxysporum* Fo2, and *F. proliferatum* ET1 protein predicted genes. It showed 10 core genes. Besides, 6241, 8356, 8033, 8439, and 10,756 specific genes were found in the AQZ-20, AP-Z13, STlab-iib, Fo2, and ET1 species, respectively. The total pan genes 48,296 were found in core and pan gene analysis, whereas the total 57,168 genes and other 48,286 were dispensable. (**b**) Gene family Orthologous Venn diagram of AQZ-20, AP-Z13, STlab-iicb, Fo2 and ET1. The analysis of the gene family of *A. rasikravindrae* and AP-Z13, STlab-iicb, Fo2 and ET1 are shown in the orthology Venn diagram. Amongst them, the homologous single-copy genes number of *A. rasikravindrae*, *F. proliferatum* ET1, *A. malaysianum* STlab-iicb, *A. phaeospermum* AP-Z13, and *F. oxysporum* Fo2 was 107, 471, 49, 60, or 101, respectively. This diagram shows the five devastating fungus species’ correlation with their corresponding colors.

**Figure 7 jof-08-00255-f007:**
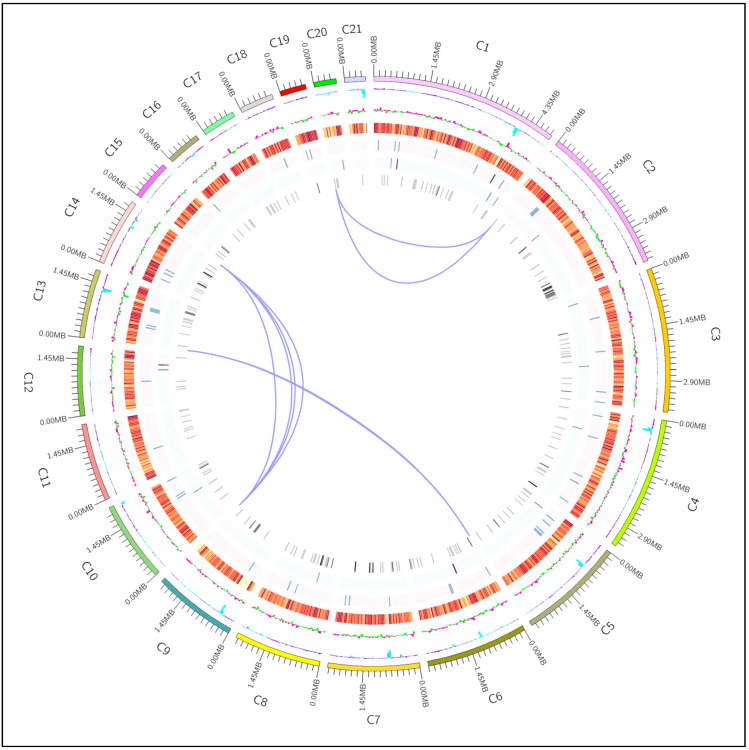
The circular whole-genome map of *A. rasikravindrae*. It is explained in the figure that from the exterior to the interior are: 1. Contigs (>1 MB in length); 2. GC content: deliberated as the % of G + C in 1 kb non-overlapping windows.

**Figure 8 jof-08-00255-f008:**
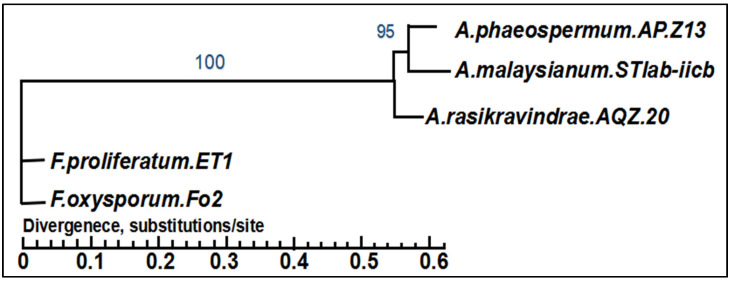
Inter-specific phylogenetic tree of *A. rasikravindrae* AQZ-20, *A. phaeospermum* AP.Z13, *A. malaysianum* STlab-iicb, *F. proliferatum* ET1, and *F. oxysporum* Fo2. As per the evaluation results of species analysis, only duplicate genes are recognized by a gene family. *A. phaeospermum*, *A. malysianum,* and *A. rasikravindrae* have shown the same evolutionary branch and have no greater evolutionary distance. *F. oxysporum* Fo2 and *F. proiferatum* ET1 have a different evolutionary branch from *A. rasikravindrae* and has greater evolutionary distance. As per the NCBI taxonomy database, *A. phaeospermum*, *A. malaysianum* STlab-iicb, *A. rasikravindrae* and *F. proliferatum* ET1, and *F. oxysporum* Fo2, belong to Xylariales, so, these are very closely linked to *A. rasikravindrae*.

**Table 1 jof-08-00255-t001:** Gene assembly data of *A. rasikravindrae* genome.

*Arthrinium rasikravindrae*	Characteristic
Contigs	32
Max_Length (bp)	4,831,660
N50_Length (bp)	2,356,432
Total length (bp)	45,874,955
GC (%)	52.66
Genome size (bp)	45,874,955
Gene number (#)	11,101
Gene total length (bp)	14,554,907
Gene average length (bp)	1311
Gene length/Genome (%)	31.73

**Table 2 jof-08-00255-t002:** *A. rasikravindrae* genome data related dispersed repeat sequences (DRs).

Type	Number (#)	Entire Length (bp)	In Genome (%)	Average Length (bp)
LTR	834	64,221	0.14	81
DNA	622	41,435	0.0903	74
LINE	459	37,226	0.0811	89
SINE	72	6894	0.015	96
RC	42	3277	0.0071	78
Unknown	12	853	0.0019	71
Total	2041	139,096	0.3032	81

**Table 3 jof-08-00255-t003:** *A. rasikravindrae* genome data related tandem repeat sequences (TRs).

Type	Number (#)	Repeat Size (bp)	Entire Length (bp)	In Genome (%)
TR	6442	1~456	282,654	0.6161
Mini-satellite DNA	4596	10~60	199,859	0.4357
Micro-satellite DNA	930	2~6	39,132	0.0853

**Table 4 jof-08-00255-t004:** *A. rasikravindrae* genome data related RNAs.

Type	Number (#)	Average Length (bp)	Entire Length (bp)
tRNA	268	87	23,415
5 s (denovo)	54	116	6243
5.8 s (denovo)	0	0	0
18 s (denovo)	1	1795	1795
28 s (denovo)	1	3340	3340
sRNA	2	236	473
snRNA	15	159	2388
miRNA	1	95	95

**Table 5 jof-08-00255-t005:** Gene product name and functions of DFVF 10 genes.

Gene ID/	Functions
A03535	Unknown
A11076	CATALYTIC ACTIVITY: ATP + a protein = ADP + a phosphoprotein; SIMILARITY: Contains 1 protein kinase domain.
A10939	Tubulin is the major constituent of microtubules. It binds two moles of GTP, one at an exchangeable site on the beta chain and one at a non-exchangeable site on the alpha-chain (By similarity); SUBUNIT: Dimer of alpha and beta chains (By similarity); SIMILARITY: Belongs to the tubulin family.
A02329	Guanine nucleotide-binding proteins (G proteins) are involved as a modulator or transducers in various transmembrane signaling systems. The beta and gamma chains are required for the GTPase activity, for replacement of GDP by GTP, and for G protein- effector interaction; SUBUNIT: G proteins are composed of 3 units, alpha, beta, and gamma; SIMILARITY: Belongs to the WD repeat G protein beta family; SIMILARITY: Contains 7 WD repeats.
A08957	Unknown function
A08380	Unknown function
A00946	Mitogen-activated protein kinase is involved in a signal transduction pathway that is activated by changes in the osmolarity of the extracellular environment. Controls osmotic regulation of transcription of target genes (By similarity). Involved in the virulence and conidia formation. Mediates tannic acid-induced laccase expression and cryparin expression; CATALYTIC ACTIVITY: ATP + a protein = ADP + a phosphoprotein; COFACTOR: Magnesium (By similarity); ENZYME REGULATION: Activated by tyrosine and threonine phosphorylation (By similarity). Hypoviruses like CHV1-EP713 induce inactivation by lowering the degree of phosphorylation in response to various environmental stresses; SUBCELLULAR LOCATION: Cytoplasm (By similarity). Nucleus (By similarity); DOMAIN: The TXY motif contains the threonine and tyrosine residues whose phosphorylation activates the MAP kinases; PTM: Dually phosphorylated on Thr-171 and Tyr-173, which activates the enzyme (By similarity). Phosphorylated in response of osmotic stress; SIMILARITY: Belongs to the protein kinase superfamily. Ser/Thr protein kinase family. MAP kinase subfamily. HOG1 sub-subfamily.; SIMILARITY: Contains 1 protein kinase domain.
A08495	Unknown function
A07052	Unknown function
A08159	Unknown function

## Data Availability

The whole-genome sequence for *Arthrinium rasikravindrae* strain AQZ-20 data has been submitted to the database of GenBank with accession no. JACVVL000000000. The assembly and sequenced genome raw data reported in this paper are associated with NCBI BioProject: PRJNA661692 and BioSample: SAMN16069862 within GenBank. The SRA accession number is SRR12768822. The authors state that the current study existing the necessary data for the conclusion and Appendix A will be fully exposed.

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
