# Peer review of "The First Whole Genome Sequence Discovery of the Devastating Fungus *Arthrinium rasikravindrae"

_jof, 2022, doi:10.3390/jof8030255_

Round 1
Reviewer 1 Report
My opinion is that the work is generally good, I think it is of interest for people who work with species close to the one studied, but I think there are a few things that need to be improved.
In the introduction of the work, confusion can be generated with the term endophyte, since this term is used for organisms that grow inside the plants without producing disease symptoms. It is true that isolated organisms such as endophytes can change to pathogenic habits depending on some factors or be pathogenic to other species other than their original host.
The isolation of the "endophyte" fungus is not sufficiently described. Example, appearance of the plants, surface disinfection protocol or controls to determine if it is really an endophyte. It is only mentioned that the sample was provided by an associate. It is probable that everything mentioned in this paragraph and the previous one is not necessary if the term endophyte is omitted, otherwise the affirmations must be supported.
I believe that the discussion can be improved by emphasizing more the genes associated with pathogenicity as some of the enzymes and mentioning more details of the secondary metabolites comparing them with similar works.
The work has some misspellings and missing subscripts (L52, L241 and L81). Figure 9 does not have italics in the name of the species.
Author Response
Comments and Suggestions for Authors
My opinion is that the work is generally good, I think it is of interest for people who work with species close to the one studied, but I think there are a few things that need to be improved.
Answer: Thank you so much for your careful review and valuable suggestions. We have thoroughly revised the manuscript as suggested.
In the introduction of the work, confusion can be generated with the term endophyte, since this term is used for organisms that grow inside the plants without producing disease symptoms. It is true that isolated organisms such as endophytes can change to pathogenic habits depending on some factors or be pathogenic to other species other than their original host. The isolation of the "endophyte" fungus is not sufficiently described. Example, appearance of the plants, surface disinfection protocol or controls to determine if it is really an endophyte. It is only mentioned that the sample was provided by an associate. It is probable that everything mentioned in this paragraph and the previous one is not necessary if the term endophyte is omitted, otherwise the affirmations must be supported.
Answer: Thank you so much for your comments and suggestions. We have removed the word "endophyte" as suggested.
I believe that the discussion can be improved by emphasizing more the genes associated with pathogenicity as some of the enzymes and mentioning more details of the secondary metabolites comparing them with similar works.
Answer: Tank you. We have added the more information accordingly.
The work has some misspellings and missing subscripts (L52, L241 and L81). Figure 9 does not have italics in the name of the species.
Answer: Misspellings and missing subscripts problem has been solved (Line 76-79), (Line 248) and (Line 52) are the corrected version. Moreover, species names have been correctly written now in italics as suggested for Figure 9, which has now renamed as Figure 8.
Reviewer 2 Report
pg 1 line 40: Arthrinium, Arthrinium rasikravindrae or Arthrinium rasikravindrii and..... why two times Arthrinium, kindly check the sentence and write Arthrinium rasikravindrae as A. rasikravindrae.
Figures 1. (a) and b are not clear, it should be clear as nothing can understand from these blurred figures.
same with Figure 2. (a) and b, kindly upload a clear version of the figures.
Figure 3. (a) and b are also not clear.
All figures in the manuscript should be more clear.
Author Response
Comments and Suggestions for Authors
Thank you so much for your careful review and valuable suggestions. We have thoroughly revised the manuscript as suggested.
pg 1 line 40: Arthrinium, Arthrinium rasikravindrae or Arthrinium rasikravindrii and why two times Arthrinium, kindly check the sentence and write Arthrinium rasikravindrae as A. rasikravindrae.
Answer: Here the first Arthrinium describes about the genus name while the second Arthrinium describes about the specie name with genus. Moreover, we have modified the specific sentence according to your suggestions. (Line 40 corrected version).
Figures 1. (a) and b are not clear, it should be clear as nothing can understand from these blurred figures.
Answer: Figure (1a) and (1b) has been improved. But now Fig 1a has been named as Figure 1 and Figure 1b has been named as Figure 2.
same with Figure 2. (a) and b, kindly upload a clear version of the figures.
Answer: A clear version of the Figure 2. (a) and b has been uploaded. But now Fig 2a has been named as Figure 3 and Figure 2b has been named as Supplementary Figure S1.
Figure 3. (a) and b are also not clear.
Answer: A clear version of Figure 3. (a) and b has been uploaded.But now Fig 3a has been named as Supplementary Figure S2 and Figure 3b has been named as Figure 4.
All figures in the manuscript should be more clear.
Answer: All the figures in the manuscript have been improved.
Reviewer 3 Report
Dear authors
Thank you very much for submitting this manuscript to the Journal of Fungi.
Here are my comments:
General comments:
The manuscript describes the outcomes of the whole genome analysis of the endophytic fungi Arthrinium rasikravindrae. While the details are well covered, the manuscript is still not very accessible for a wider readership. In the comments blow, I try to highlight ways to improve this. I would also like to invite the authors to decide which information could be moved to supplementary materials as the manuscript would benefit from a focus on the most important outcomes.
Please ensure that in future submissions the figures are readable – currently they are too small.
Specific comments
Line 2-3 a more gripping title may be “The whole Genome of the endophytic and devastating fungus Arthrinium rasikravindrae”
L21 are 4 ascomycestes as well endophytic?
L88-91: please use full names e.g. Ustilago maydis
L92-93 sentence unclear
Table 1 could go to supplementary materials – there seems to be no reference to table 1 in the main text
L177 say here that the material was isolated from bamboo shoots
L180 please explain N50 contig distance to the reader
L214 please introduce the KEGG database to the reader (same applies to all databases & platforms!)
L266 please explain why it is important to know about the P450 proteins (this section is difficult to follow)
L280 please explain the M9 format and why it is important
There seems to be no info in the text about the data shown in Fig 3b
L301 section 3.7 needs an introduction and more information as currently too short
L307-308 please explain T1PKS & NRPS – general comment: I would not give the numbers in brackets e.g “In addition, 13 genes and one cluster…” (applies to all numbers in brackets)
L313 explain AA, CBM, CE, GH, GT, PL
L321 what is “unaffected pathology” and “reduced virulence”? Please explain.
General comment: please give the names & functions of the gene products like A03535 as far as this is known. If this is not known, please say gene product of unknown function.
L327 why are the genes CAM and GzGPA1 Important here?
L334 what are virulence issues?
L338 why is F oxysporum of interest here?
There doesn´t seem to be a reference to Fig 6 in the main text
I would put more emphasis on the information shown in Fig 7 and Fig 8 in the manuscript as this is the interesting bit.
L382 please define core and pan genes
L390 please introduce why the cluster analysis was performed
L404-407: please say here that these are the 10 common genes shown in Fig 8A (again please include gene product names and functions – a table may work here better
L423-435: this section of the discussion is redundant with earlier information n the manuscript.
General comment: the Discssion should focus more on e.g. the 10 common genes as well as the key similarities and differences to the closely related genomes & species
Author Response
Comments and Suggestions for Authors
Dear authors
Thank you very much for submitting this manuscript to the Journal of Fungi.
Answer: Thank you so much for your careful review and valuable suggestions. We have thoroughly revised the manuscript as suggested.
Here are my comments:
General comments:
The manuscript describes the outcomes of the whole genome analysis of the endophytic fungi Arthrinium rasikravindrae. While the details are well covered, the manuscript is still not very accessible for a wider readership. In the comments blow, I try to highlight ways to improve this. I would also like to invite the authors to decide which information could be moved to supplementary materials as the manuscript would benefit from a focus on the most important outcomes.
Answer: Thanks for your valuable comments. Table 1., Supplementary Figure S1.NR annotation of species, Supplementary Figure S2. TCBD functional classification, Supplementary Figure S3. Carbohydrate enzyme classification and annotation, Supplementary Figure S4. Secondary metabolic gene Cluster. Supplementary Figure S5. PHI phenotype classification. have been added to the supplementary materials as you suggested.
Please ensure that in future submissions the figures are readable – currently they are too small.
Answer: Modified as suggested.
Specific comments
Line 2-3 a more gripping title may be “The whole Genome of the endophytic and devastating fungus Arthrinium rasikravindrae”
Answer: The title of the manuscript has been changed as advised. (Line 2-3 corrected version).
L21 are 4 ascomycestes as well endophytic?
Answer: yes, 4 Ascomycetes mentioned in this research are endophytic as well (Line- 20 corrected version).
L88-91: please use full names e.g. Ustilago maydis
Answer: Modified as suggested (Line 87-91 corrected version).
L92-93 sentence unclear
Answer: Modified as suggested (Line 92-94 corrected version).
Table 1 could go to supplementary materials – there seems to be no reference to table 1 in the main text
Answer: Table 1 has been shifted to the supplementary material.
L177 say here that the material was isolated from bamboo shoots
Answer: Improved (176 corrected version).
L180 please explain N50 contig distance to the reader
Answer: The word ‘Distance’ has been replaced with the correct word ‘ Length’. (Line 180 corrected version).
L214 please introduce the KEGG database to the reader (same applies to all databases & platforms!)
Answer: All the Databases and Platforms throughout the paper has been provided. Line 203-208 corrected version)
L266 please explain why it is important to know about the P450 proteins (this section is difficult to follow)
Answer: Cytochrome P450 (CYP) is a hemeprotein that plays a key role in the metabolism of drugs and other xenobiotics. Please refer to line 270-289.
Importance: CYP is a complex and important component of drug metabolism. It is the root of many drug interactions due to inhibition, induction, and competition for common enzymatic pathways by different drugs. Genetic variability of CYP is also a significant source of unpredictable drug effects.
L280 please explain the M9 format and why it is important
Answer: Thanks for pointing, this is M8 rather than M9. The BLAST m8 format has become a de facto standard for blast model as it is quite straightforward and is widely used by sequence alignment programs to summarize results.
Li, G., Wei, H., Bi, J., Ding, X., Li, L., Xu, S., ... & Ren, W. (2020). Insights into dietary switch in cetaceans: evidence from molecular evolution of proteinases and lipases. Journal of Molecular Evolution, 88(6), 521-535.
There seems to be no info in the text about the data shown in Fig 3b
Answer: The information related Fig.3b which has been renamed as Fig.4 is available in the results section.
L301 section 3.7 needs an introduction and more information as currently too short
Answer: More information about section 3.7 has been added.
L307-308 please explain T1PKS & NRPS – general comment: I would not give the numbers in brackets e.g “In addition, 13 genes and one cluster…” (applies to all numbers in brackets)
Answer: NRPS, Non-Ribosomal Peptide Synthetase Cluster; T1PKS, Type I Pks (PolyKetide Synthase). Modified as suggested. (Line 311-317 corrected version).
L313 explain AA, CBM, CE, GH, GT, PL
Answer: The following abbreviations have been now written with complete names which can make them easy for the reader to understand. E.g. Auxiliary Activities (AA), carbohydrate-binding module (CBM), carbohydrate esterases (CE), GH (glycoside hydrolases), glycosyltransferases (GT), and polysaccharide lyase (PL). (line 298-301 corrected version).
Moreover the figure S3 and S4 have been shifted to supplementary file.
L321 what is “unaffected pathology” and “reduced virulence”? Please explain.
Response: Sorry for ambiguous statements, unaffected pathogenicity and reduced virulence are functional class obtained through PHI database. We have revised the sentence in line 321-323 in the revised manuscript.
General comment: please give the names & functions of the gene products like A03535 as far as this is known. If this is not known, please say gene product of unknown function.
Answer: Gene product of unknown function.
L327 why are the genes CAM and GzGPA1 Important here?
Answer: A03535 (I1RNF9) genes in A. rasikravindrae were paired with the CAM gene of M. oryzae, which caused rice blast and head blast in host Triticum aestivum [76], and the GzGPA1 gene of Fusarium graminearum, which caused Fusarium ear blight [77]. Therefore, the A02487 and A03535 genes might be associated with rice blast pathogenicity [9], and Fusarium ear blight in Barley. The GzGPA1 pathogenicity of the barley-directed strain was the same as that of the wild-type strain. GzGPA1 is a conserved low salt protein that encodes 353 (AA), highly similar to A. nidulans FadA in the genome database of F. graminearum (93% identity) and also negatively controls mycotoxin production. (line 525-533). These evidences provide relevant proof about the importance of CAM and GzGPA1 genes. The references used in the answer are present in my article with the same numbering for confirmation.
L334 what are virulence issues?
Answer: Virulence issues can be defined as a wide range of molecules produced by pathogenic microbes that enhance their ability to evade their host defenses and cause disease.
L338 why is F oxysporum of interest here?
Answer: 1. The pathogen and the target pathogen are both semi-known, and the occurrence rate is high in bamboo branches, both in the same ecological environment.
- Moreover, in comparative genomic analysis, database found that the genetic relationship between them is very close.
There doesn´t seem to be a reference to Fig 6 in the main text
Answer: Figure 6 has been renamed as Figure 5 and main text added. Please refer to section 3.12.
I would put more emphasis on the information shown in Fig 7 and Fig 8 in the manuscript as this is the interesting bit.
Answer: Figure 7 and 8 have been renamed as Figure 6 and 7. More information related figure 6 has been added in the manuscript. Please refer to 6 captions. Moreover, we have improved the quality of Figure 6 and 7, to make it more convenient for the readers to understand.
L382 please define core and pan genes
Response: Thanks for pointing. we added a small description of core and pan genes in section 3.13.2. We did not add detailed explanation as it is a common terminology for genome analysis articles.
L390 please introduce why the cluster analysis was performed
Answer: We have added the required details in section 3.13.3. The gene families of A. rasikravindrae, A. phaeospermum, A. malaysianum, F. oxysporum, and F. proliferatum were subjected to cluster analysis. Cluster analysis was performed to find the unique genes family of core and pan-genome. The total and unique number of gene families of A. rasikravindrae and other spp. were presented in the form of Venn diagrams.
L404-407: please say here that these are the 10 common genes shown in Fig 8A (again please include gene product names and functions – a table may work here better
Answer: Thanks for suggestion, we agreed with your idea. Table 6 has been added. We have also added the data of core genes in Table 5 in revised version. According to your suggestion we added the about 10 common genes line 424-425.
L423-435: this section of the discussion is redundant with earlier information n the manuscript.
Answer: Thanks for suggestion, we have removed redundant ideas from discussion.
General comment: the Discussion should focus more on e.g. the 10 common genes as well as the key similarities and differences to the closely related genomes & species.
Answer:10 common genes related information has been provided in the discussion part (line 519-520 revised corrected version). while all the information regarding genomes and species are mentioned in Supplementary table S1. In our previous research on the whole geneome sequence of Arthrinium pheaospermum and whole geneome sequence of Diphorthe capsici by Li et al 2021 and Fang et al 2021 was done and published recently, whose studies are quite similar to the the one we conducted in this experiment.
Round 2
Reviewer 2 Report
Accept